# clusTransition: An R package for monitoring transition in cluster solutions of temporal datasets

Muhammad Atif[1,2]*, Friedrich Leisch[2]

**1** Department of Statistics, University of Peshawar, Peshawar, Pakistan, **2** University of Natural Resources and Life Sciences, Vienna, Austria

* dratif96@gmail.com

## Abstract

Clustering analysis' primary purpose is to divide a dataset into a finite number of segments based on the similarities between items. In recent years, a significant amount of study has focused on the spatio-temporal aspects of clustering. However, clusters are no longer regarded as static objects since changes influence them in the underlying population. This paper describes an R package implementing the MONIC framework for tracing the evolution of clusters extracted from temporal datasets. The name of the package is clusTransition, which stands for Cluster Transition. The algorithm is based on re-clustering cumulative datasets that evolve at successive time-points and monitoring the transitions experienced by the clusters in these clustering solutions. This paper's contribution is to demonstrate how the package clusTransition is developed in the R programming language, and its workflow is discussed using hypothetical and real-life datasets.

## 1 Introduction

The prime goal of clustering analysis is the organization of a dataset into a finite number of segments according to the similarities within objects. Ideally, the set of objects in the same segment should be comparably similar to one another than to the objects belonging to different partitions [1]. Each individual partition is known as a cluster, whereas the objects belonging to the same cluster are called its members [2, 3]. Its applications covers many real-world applications, ranging from business and economics, marketing, pattern recognition, medical sciences, image processing to big data analysis [4]. For example, in the field of market segmentation, better marketing strategies can be adopted by clustering the customers with similar demographic or buying characteristics [5]. In a similar passion, clustering might be helpful in better understanding the disease and targeting appropriate treatment by subgrouping the patients into homogeneous sets based on psychological inventory scores [6]. Since the notion of clustering is not precisely described, consequently, several algorithms/ models have been proposed in the literature, and all of them may result in well different clustering solutions [7, 8].

Scale datasets: https://ess-search.nsd.no/CDW/ConceptVariables Link for household Electric Power Consumption: https://archive.ics.uci.edu/452ml/datasets/individual+household+electric+power+consumption Link for Intel Lab sensor datasets: https://www.kaggle.com/datasets/divyansh22/intel-berkeley-research-lab-sensor-data.

**Funding:** The author(s) received no specific funding for this work.

**Competing interests:** The authors have declared that no competing interests exist.

It is already true that in recent years, a considerable amount of research work conducted is based on investigating the spatio-temporal properties of clustering. In these applications, clusters are no longer considered static objects, as they are affected by changes occurring in the underlying population [9, 15]. The inclusion of new data records to the original population over time may affect the cluster's memberships, and entirely different clustering solutions may be generated at later time-points. This transition in clustering solutions may include the disappearing of a specific cluster(s), migration of some elements from one cluster to another, splitting of a cluster into several, several clusters splicing together to form one, survival of a cluster and emerging of new ones. The survived clusters can experience internal transition, including changes in location, size, and density [10, 11]. Various topics such as spatio-temporal, evolutionary, stream, and incremental clustering address this issue by adopting the dataset that changes over time. Tracing and understanding the phenomena behind this transition is of practical importance for effective decision-making. This can be helpful in various fields like marketing, fraud detection, networking, scientific publication, health, etc. [12].

In many real-world applications, clustering of the data stream is performed all time to identify the changes occurring in the pattern of the underlying phenomena [13]. Since in a stream, new data items are continually generated, which join the underlying population at a regular interval. Therefore, in order to control part of the data that contributes to the pattern in data mining, the stream needs to be discretized into subsets based on some attributes that have an order. This data discretization into subsets is called the windowing approach and is mainly done based on time. Some of the most commonly used examples are landmark, sliding, and damped window models [14]. These models are discussed in the next section.

[15] introduces the notion of evolutionary clustering to process the times-tamped dataset by producing a sequence of clustering solutions. That is, a clustering solution for each time-step of the temporal data. The algorithm optimizes two competing criteria i.e. each clustering in the sequence should be similar to the clustering at the previous time-step, while at the same time should accurately reflect the data arriving during that time-step. This framework is further extended to spectral clustering [16], density-based clustering [17], and Hierarchical Dirichlet Process with the Hidden Markov model [18].

Using a totally online method, Hyde et al. [19] offer an algorithm that clusters the evolving data streams into arbitrary shaped clusters. The approach consists of two stages: the first stage finds micro-clusters in the datasets, and the second step merges these micro-clusters into macro-clusters. In a similar vein, Fahy et al. [20] describe an Ant Colony Stream Clustering technique built on a density-based methodology that recognises clusters as a collection of micro-clusters. To read a stream and create micro-clusters in the window pane, the method uses a tumbling window model. By combining the related clusters based on a similarity index, these clusters are then further refined. Fahy and Yang [21] further enhance this technique to address the multi-density issue in the density-based clustering strategy. This method uses the local radius of each cluster to identify clusters, and it then tracks changes in the solutions. For the first time, multiple view clustering challenges are addressed by Huang et al. [22] in MVStream clustering method. In order to assign cluster labels to the data items that include summary statistics, this technique creates support vectors from various views of the data objects. Similarly, some studies have been conducted for measuring the similarities between the trajectory in the dynamic environment [23–25].

## 2 Window models

In a landmark window model, all items that arrive after some specific time-point (landmark time) are maintained and cannot be discarded irrespective of window size. The window size is uncontrolled and keeps increasing as time progresses [26, 27]. The data records that arrives in the interval $(t_{i-1}, t_i)$ are accumulated according to the equation given by:

$$D_t = \bigcup_{i=1}^{t} d_i, i = 1, 2, ..., n \tag{1}$$

where $n$ is the number of time-points and $t$ is the current time-point. Implementation of the landmark window model will generate $n$ window panes, where each pane contain data items evolving from starting time-point $t_1$ to the current time-point $t_i$.

The sliding window model, on the other hand, is based on a fixed size of window $w$ that contains only those objects falling in the interval $[t_i - w + 1, t_i]$, while older cases are discarded. In such type of model, as time progress, the window slides forward while keeping its size $w$ by including new data records and discarding the older ones [27, 28]. The scenario of the sliding window model can be described in the equation below:

$$D_1 = d_1 \tag{2}$$

$$D_2 = \bigcup_{i=1}^{w} d_i \tag{3}$$

$$D_3 = \bigcup_{i=2}^{w+1} d_i \tag{4}$$

$$\ldots$$

$$D_m = \bigcup_{i=n-w+1}^{n} d_i \tag{5}$$

where $m$ is the number of window panes and is equal to $n - w + 2$, $n$ is the number of time-points, and $w$ is the sliding window size.

## 3 The change detection algorithm

In order to monitor and trace the evaluation of clusters extracted from re-clustering of cumulative datasets [29] introduced a framework known as 'MONIC' algorithm. This algorithm is based on clustering cumulative datasets arriving at discrete time-points $t_1, t_2, \ldots, t_n$. Initially, the data is collected at time-point $t_1$, and as time progresses new data records join the data set at regular interval of time. The initial datasets $d_1, d_2, \ldots, d_n$, are accumulated and re-clustered at each time-point $t_1, t_2, \ldots, t_n$ to monitor and detect the cluster evolution over time.

The algorithm is mainly based on the idea of a non-symmetric overlap matrix between two clustering extracted from cumulative datasets at two different time-points. Let $\xi_i = \{X_1, \ X_2, \ \ldots, \ X_{k_1}\}$ be a set of clusters extracted from dataset $D_i$ at time point $t_i$ and is referred to as first clustering. Similarly, let $\xi_j = \{Y_1, \ Y_2, \ \ldots, \ Y_{k_2}\}$ be a set of clusters

extracted from dataset $D_j$ at time point $t_j$ ($i<j$) and is referred to as second clustering. Then the overlap matrix can be defined as:

$$overlap\left(X_i, \ Y_j\right) = \frac{|X_i \cap Y_j|}{|X_i|} \quad i = 1, 2, \cdots, k_1, j = 1, 2, \cdots, k_2 \qquad (6)$$

where $k_1$ is the number of clusters from the first clustering $\xi_i$, and $k_2$ is the number of clusters from second clustering $\xi_j$. This will generate a matrix of order $k_1{}^*k_2$, where rows and columns describe first and second clustering respectively. The value on the corresponding element of the matrix represents the similarity index between cluster $X_i$ and $Y_j$. The MONIC framework assumes hard clustering where each observation belongs to one and only one cluster [30].

In the context of this algorithm, the transition is the change experienced by a cluster $X_i \epsilon \xi_i$, when it has been perceived at second clustering $\xi_j$. This change in the clustering solution is referred to as an external or internal transition. External transition concern the relationship of cluster found at clustering $\xi_i$ to the clusters found at clustering $\xi_j$, whereas internal transition is regarded as changes that occurred in the structure of the survived clusters.

The external transition is categorized into five categories i.e. Survive, Merge, Split, Disappear, and Emerge candidates. The cluster $X_l \epsilon \xi_i$ may survive into $Y_m \epsilon \xi_j$, clusters $\{X_{l_1}, \ X_{l_2}\} \in \ \xi_i$ may merge to form $Y_m \epsilon \xi_j$, or cluster $X_l \epsilon \xi_i$ may split into various daughter clusters $\{Y_{m_1}, \ Y_{m_2}\} \in \ \xi_j$. If a cluster $X_l \epsilon \xi_i$ does not experience any of the above transitions, then it disappears. Similarly, if a cluster $Y_m \epsilon \xi_j$ is not a result of any external transition from its ancestors, then it is a newly emerged candidate. The *overlap* between $X_l \epsilon \xi_i$ and $Y_m \epsilon \xi_j$ serve as an indicator of identifying the external transition experienced by clusters at clustering $\xi_i$. This value is compared with a minimum threshold value say $\tau \epsilon [0.5, 1]$ to identify match of $X \epsilon \xi_i$ in $Y \epsilon \xi_j$. A cluster $X_l \epsilon \xi_i$ is said to survive in $Y_m \epsilon \xi_j$ if this is the only cluster that has an *overlap* of greater than $\tau_{survive}$. If at least two clusters from $X \epsilon \xi_i$ (such as $X_{l_1}$ and $X_{l_2}$ have an overlap of greater than $\tau_{survive}$ with $Y_m \epsilon \xi_j$), then it is a case of merge i.e. $X_1$ and $X_2$ merge to form $Y_m$. Furthermore, a cluster is said to split in daughter clusters, if the overlap of $X_l$ with $Y_{m_1}$ and $Y_{m_2}$ is greater than $\tau_{split}$ and collectively their overlap is greater than $\tau_{survive}$, i.e. for split the following two conditions are required.

$$Overlap(X_l, \ Y_m) > \tau_{split} \quad m = 1, 2, ...M \qquad (7)$$

$$\sum_{m=1}^{M} Overlap(X_l, \ Y_m) > \tau_{survive} \qquad (8)$$

where M is the number of daughter clusters from second clustering.

The *overlap* can not be used as an indicator for monitoring the changes in the form of survived clusters. The shift in the location of the survived cluster ($X_l \rightarrow Y_m$) can be traced by calculating Euclidean distance between their centroids normalized by the minimum radius. This information can be summarized in the following formula:

$$location.difference = \frac{d(\bar{X}_l, \ \bar{Y}_m)}{min\left(r_X, \ r_Y\right)} \qquad (9)$$

where $\bar{X}_l$ and $\bar{Y}_m$ are the centroids of clusters $X_l$ and $Y_m$ respectively, and $d(\bar{X}_l, \ \bar{Y}_m)$ is the Euclidean distance between them. The $r$ denotes radius of the corresponding clusters and is

computed as the maximum distance of an object from its cluster centroid. If the absolute value of *location.difference* is greater than $\tau_{location}$, then the algorithm will detect a shift in location of the survived cluster.

For density transition, the average distance of objects from cluster centroid can be computed. The formula for the density of cluster is given by:

$$avgDistance = \frac{1}{|X_l|} \sum_{i=1}^{n_l} (X_{li} - \bar{X}_l) \tag{10}$$

The difference in density of cluster $X_l$ survived in $Y_m$ is normalized by the minimum radius i.e.

$$density.difference = \frac{avgDistance_X - avgDistance_Y}{min\,(r_X,\ r_Y)} \tag{11}$$

If the absolute value *density.difference* is less than $\tau_{density}$ then there is no change in density of the survived cluster. On the other hand, if the absolute value is greater than $\tau_{density}$ then a change in density would be detected. If *density.difference* is positive then the cluster is more compact than its ancestors, otherwise, it becomes more diffuse.

## 4 Package description

The state-of-the-art "MONIC" algorithm is implemented in the **R-software** via package **clusTransition**. The package can be used for tracing and monitoring the evolution of clustering solutions in cumulative datasets over time. In this section, we briefly describe the functions and methods exported by the package in detail. Fig 1 below demonstrates the workflow of the package.

Table 1 below summarizes the functions, methods, and classes exported by the package along with its corresponding arguments and slots.

More details about these functions and classes are described below.

### 4.1 Function *Transition()*

The evolution of clusters can be traced using the primary function *Transition()*, which exports an object of class S4. In implementing the package **clusTransition**, we have considered the portability of the functions for various types of hard clustering algorithms. A typical call to the *Transition()* function involves three essential pieces: the data input (**listdata, listclus, overlap**), choice of window `swSize`, and the threshold parameters. The user must only provide the `swSize` and `k` arguments in case of importing datasets using the `listdata` argument. This function has the following interface:

> *Transition(listdata, listclus = NULL, Overlap = NULL, swSize = 1, typeind = 1,*
> + *survival_thresHold = 0.8, split_thresHold = 0.3, location_thresHold = 0.3,*
> + *density_thresHold = 0.3, k)*

We took into account the portability of the functions for many kinds of hard clustering algorithms while developing the clusTransition package. For this purpose three different options i.e. `listdata`, `listclus`, and `Overlap` are provided for importing the data.

The `listdata` imports the raw data stream at discrete time points $t_1, t_2, \ldots, t_n$. A sequence of cluster solutions are generated from the stream using *k-means* clustering

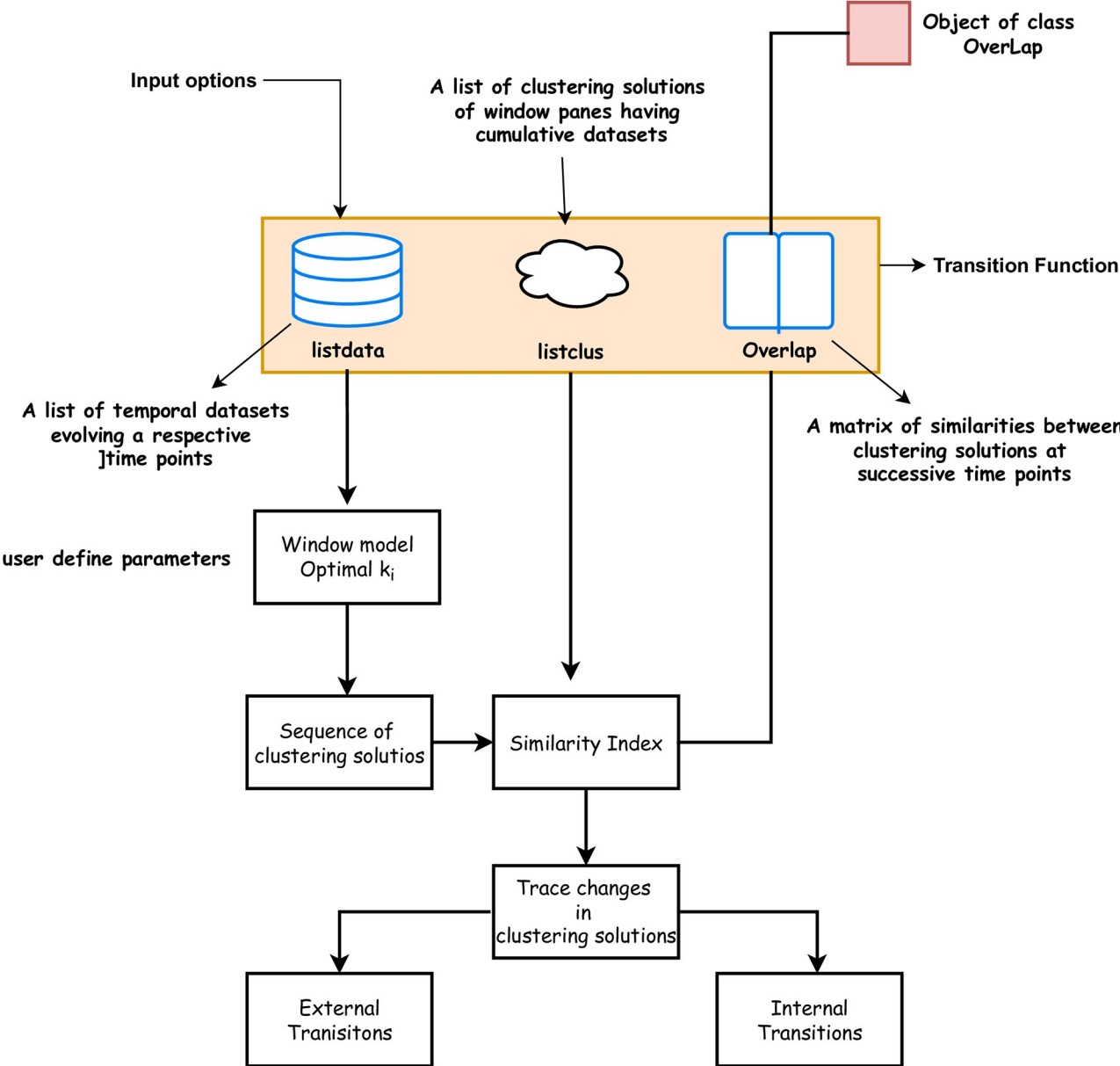

**Fig 1. Workflow diagram of the package.** The Transition function exported by the package offers three different options for importing datasets. The function then trace changes in clustering solutions.

**Table 1. Functions, methods and classes exported by the package *clusTransition*.**

| Name | Type | Description |
|---|---|---|
| *Monitor cluster evolution* Transition(listdata, . . .) | Function | Implements the change detection algorithm and trace the evolution of clusters over time. Return an object of class MONIC. |
| *OverLap class* new("OverLap") | S4 class | Class containing cluster representatives and overlap matrices |
| *Overlap* Overlap(object, e1, e2) | S4 method | Method for initializing slots OverLap class |
| *plot cluster evolution* moplot (object) | Function | plot the MONIC class |

algorithm. Each element of the list corresponds to the dataset at a single time point. The number of clusters in each accumulative data matrix is specified by the argument **k**.

On the other hand, the `listclus` argument imports the clustering solutions at successive time-points to allow clusters other than *k*-means. Each element of `listclus` is a nested list that contain clustering solutions at corresponding time point i.e. $\xi_i = \{X_1, X_2, \ldots, X_{k_i}\}$.

`Overlap` is a List of numeric matrices containing similarity measures between clusters extracted at consecutive time points. The similarity between clusters are computed using Eq 6. The `Overlap` method exported by the package can be used to compute the similarity matrices.

`swSize` indicates size of the sliding window model. The default value of `swSize = 1` implements the landmark window model and discretize the stream according to Eq 1. Whereas other numeric values discretize the stream using a sliding window scenario according to Eq 5. The sliding window size can only be provided if `listdata` argument is chosen.

The `survival_thresHold`, `split_thresHold`, `location_thresHold`, and `density_thresHold` are minimum threshold value for survival of clusters from $X \epsilon \xi_i$ to $Y \epsilon \xi_j$, split of cluster $X \epsilon \xi_i$ to $\{Y_{m1}, Y_{m2}\} \epsilon \xi_j$, shift in location, and changes in density of survived clusters respectively. These are user defined parameters and belongs to the interval (0,1).

One of the most perplexing problems with most clustering algorithms is deciding the ideal number of partitions. This is a crucial parameter for partitioning, hierarchical and model-based clustering algorithms. The number of clusters one wants to generate from a dataset has to be predefined. There are several ways of estimating the optimal number of clusters *k*, such as the silhouette, Gap, and Elbow methods. k is a numeric vector containing the relevant number of clusters at the corresponding time-point. The length of *k* is to be determined from the `swSize`. This argument should only be provided if **listdata** argument is chosen.

Typing the object's name comprising the *Transition()* function's output will produce external and internal transition results at each time point. External transition includes the number of clusters still existent, absorbed by others, split into various, disappeared and newly emerged at second clustering. Internal transition comprises changes in the location and density of the survived clusters.

Along with this information, the **Monic** object holds the cluster's radius, membership, and distance between cluster centres.

## 4.2 *OverLap* class

This is an object of class `OverLap` that contains summaries of first and second clustering. This object has eight slots that work as input for tracking the evolution of clusters by the *Transition()* function. The slots include a numeric matrix containing the similarities between clusters generated at first and second clustering (Overlap computed from Eq 6), the cluster's membership vector, radius, centres, and an average distance of items from the cluster's centres (computed from Eq 10). In addition, this has the following interface:

>*obj <- new("OverLap")*

## 4.3 *Overlap* method

This method initializes the slots of an object having class **OverLap** by importing the clustering solution ξ of cumulative datasets *D* at two consecutive time points *i* and *j*. Clusters at each data

point should be provided as a list of matrices, where each matrix contains a data set belonging to one cluster. It has the following interface.

> $Overlap <- Overlap(object, e1 = C1, e2 = C2)$

where $\mathtt{e1}$ is the set of clusters $\xi_i = X_1, \; X_2, \; ..., \; X_{k_1}$ obtained at time point $t_i$ from cumulative dataset $D_i$, $\mathtt{e2}$ is the set of clusters $\xi_j = Y_1, \; Y_2, \; ..., \; Y_{k_2}$ obtained at time point $t_j$ from cumulative dataset $D_j$, and $\mathtt{object}$ is an object of class $\mathtt{OverLap}$.

### 4.4 Function *moplot()*

This method plot 3 bar-plot and 1 line graph. The first stack bar-plot shows $\mathtt{SurvivalRatio}$ and $\mathtt{AbsorptionRatio}$, second bar-plot shows number of new emerged clusters at each time stamp, third bar-plot shows number of disappearance at each time stamp. The line graph shows $\mathtt{passforward}$ Ratio and $\mathtt{SurvivalRatio}$.

> plot(obj)

## 5 Simulation example

Let us assumes that a data stream consist of datasets $d_1, d_2, \ldots, d_n$ arriving at corresponding time-points $t_1, t_2, \ldots, t_n$ respectively. For the generation of initial dataset $d_1$, we use a generator that takes into account the number of clusters (k), size of each cluster, and separation value between theme [31]. While the generator for generating other streams like $d_2, d_3, \ldots, d_n$ consider the center of each cluster, size of each cluster, and the co-variance structure between them as input [32, 33].

As a working example, we generate a data stream sprouting at four consecutive time points. Fig 2 below demonstrates the scenario for generating datasets $d_i$, $i$ = 1, 2, 3, 4 at four time points. The new objects joining the underlying population are shown by red color whereas older records are displayed by black color.

## 6 Pre-processing

Prior to the implementation of the change detection algorithm in cluster solutions over time, the user needs to pre-specify some relevant parameters. First of all, the user needs to decide a suitable windowing approach for the accumulation of datasets evolving at successive time points. For this purpose, we offered two types of windowing approaches in the package i.e. landmark and sliding window models. Implementation of the windowing approach will accumulate the datasets at corresponding time points according to the chosen model and will generate window panes at successive time points. In the second phase, the optimal number of clusters in each window pane $D_i$ at the corresponding time point must be determined using an appropriate technique. For illustration purposes, we use worked examples based on the datasets simulated in section IV. The datasets are accumulated according to the landmark and sliding windowing approaches, and then the optimal number of clusters was estimated in each window pane $D_i$.

The implementation of the landmark window model will produce four window panes. Each pane will contain the datasets generated between $[t_1, t_i]$, where $t_i$ represent the current time point. Table 2 below demonstrates the number of objects and optimal number of clusters in each window pane $D_i$ estimated from Gap statistics at corresponding time points $t_i$.

Similarly, the implementation of a sliding window of size 3 will generate 3 window panes. Table 3 below demonstrates the number of objects and optimal number of clusters in each window pane $D_i$.

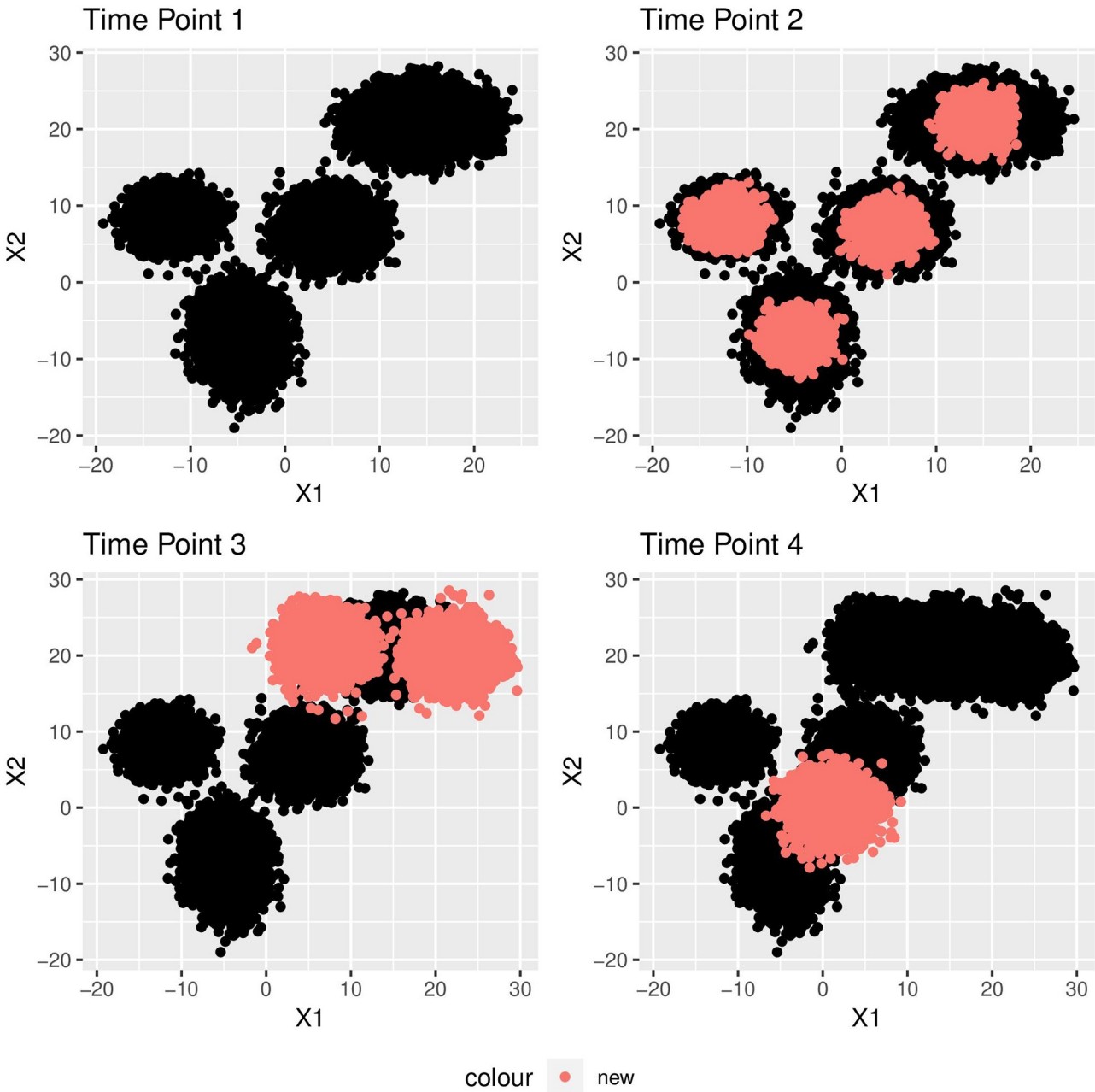

**Fig 2. Data stream generated at four discrete time stamps.** The new data items at each time stamp is shown by the red color, whereas the older data items are shown by black color.

## 7 Implementation of function *Transition()*

In this section implementation of the primary function, *Transition()* is presented using working examples. The data stream simulated in section 5 is used for monitoring the cluster evolution over time. The function provides three different options for importing the datasets, which are explained in subsections below.

**Table 2. Optimal number of clusters in landmark window model datasets.**

| Time points | $t_1$ | $t_2$ | $t_3$ | $t_4$ |
|---|---|---|---|---|
| Window panes | $D_1$ | $D_2$ | $D_3$ | $D_4$ |
| *Number of objects ($n_i$)* | 20,000 | 32,000 | 38,000 | 41,000 |
| *Number of clusters ($k_i$)* | 4 | 4 | 5 | 4 |

Table discretize the data stream according to the landmark window model explained in Eq 1. The landmark window model provide *n* window panes of cumulative datasets.

## 7.1 Looking at listdata argument

The argument `listdata` is a list of matrices or data frames containing the datasets $d_1$, $d_2, \ldots, d_n$ evolving at corresponding time-points $t_1, t_2, \ldots, t_n$. The $i^{th}$ element of the `listdata` comprises set of data items $d_i$ that evolve at corresponding time point $t_i$. At this point the *Transition()* function accumulates the datasets $d_i$ according to the suitable windowing approach provided in `swSize` argument. The default value i.e `Swsize = 1` will implement landmark window model, whereas other integer values implements sliding window model. The accumulation of datasets $d_i$ will generate window panes $D_i$ that contain cumulative datasets at successive time points. Each window pane $D_i$ will be re-clustered by using *cclust()* function from **flexclust** package [34]. The optimal number of clusters in cumulative datasets $D_i$ should be decided by the user and must be imported via argument `k` of the function. Both `k` and `swSize` arguments are used only if `listdata` option is chosen for importing datasets $d_i$. The argument `typeind = 1` allows the user to implement `listdata` argument. Monitoring and tracking the evolution of clusters using the landmark window model is shown in the example below.

**7.1.1 Example (listdata argument with landmark window model).** The default value of `swSize = 1` implements the landmark window model and generates *n* window panes of cumulative datasets $D_i$ according to Eq 1. In this working example, the datasets generated in section 5 is used. According to Table 2 in this simulated example window panes $D_1$, $D_2$, $D_3$, and $D_4$ comprises of 4, 4, 5, and 4 clusters respectively. Hence the *Transition()* function with arguments `listdata = listdata, swSize = 1, typeind = 1,` `Survival_thrHold = 0.8, Split_thrHold = 0.3,` and `k = c(4,4,5,4)` can be implemented as:

> \>*library(clusTransition)*
> \>*listdata <- list(d1, d2, d3, d4)*
> \>*clusterTrace <- Transition(listdata = listdata, swSize = 1, typeind = 1,*
> + *Survival_thrHold = 0.8, Split_thrHold = 0.3, k = c(4,4,5,4))*

**Table 3. Optimal number of clusters in sliding window model datasets.**

| Time points | $t_1$ | $t_2$ | $t_3$ |
|---|---|---|---|
| Window panes | $D_1$ | $D_2$ | $D_3$ |
| *Number of objects ($n_i$)* | 20,000 | 38,000 | 21,000 |
| *Number of clusters ($k_i$)* | 4 | 5 | 6 |

Table discretize the data stream according to the sliding window model explained in Eq 5. The landmark window model provide *n-w+2* window panes of cumulative datasets, where *w* is the size of sliding window.

This will generate two tables, displaying the number of clusters experiencing external and internal transition at successive time points. The first table in the output comprises the number of clusters that experience external transitions at corresponding time points $t_j$. Similarly, the second table comprises the number of survived clusters that undergone internal transitions at corresponding time points. Hence the full summary of external and internal transitions are shown below.

The object `clusterTrace` returned by the *Transition()* function is an object of class **S4**, named Monic. The object contains the candidates that experience external and internal transitions at successive time points. The slots ending with x represent candidates that adopt external transitions from first clustering $\xi_i$. Whereas the slots ending with y represent the candidates that evolve as a result of corresponding external transition at second clustering $\xi_j$. For example, the candidates that experience external transitions at time point $t_3$ can be retrieved as:

*clusterTrace*
*Survival Threshold = 0.8 , Split Threshold = 0.3*
*External Cluster Transition Count*

|  | *Survive* | *Merged* | *Split* | *Died* | *New.Emerged* |
|---|---|---|---|---|---|
| *Time Step.2* | *4* | *0* | *0* | *0* | *0* |
| *Time Step.3* | *4* | *0* | *1* | *0* | *0* |
| *Time Step.4* | *3* | *2* | *0* | *0* | *0* |

*Location Threshold = 0.3 , Density Threshold = 0.3*

*Internal Cluster Transition Count*

|  | *Shif* | *No.Shift* | *Compact* | *Diffuse* | *No.change.Compactness* |
|---|---|---|---|---|---|
| *Time Step.2* | *0* | *4* | *0* | *0* | *4* |
| *Time Step.3* | *0* | *4* | *0* | *1* | *3* |
| *Time Step.4* | *0* | *3* | *0* | *2* | *1* |

*Available components:*
*======================*

*"SurvivalCanx" "SurvivalCany" "SplitCanx" "SplitCany" "MergeCanx" "MergeCany"*
*"EmergCan" "ShiftLocCan" "NoShiftLocCan" "MoreCompactCan" "MoreDiffuseCan"*
*"NoChangeCompactCan" "Centersx" "Centersy" "clusterMem" "avgDisx" "avgDisy"*
*"rx" "ry" "SurvivalRatio" "AbsorptionRatio" "passforwardRatio"*

*>clusterTrace@TimeStep[[3]]@SurvivalCanx*
*[1]    1    3    4*
*>clusterTrace@TimeStep[[3]]@SurvivalCany*
*[1]    1    4    2*
*>clusterTrace@TimeStep[[3]]@SplitCanx*
*[1]    2*
*>clusterTrace@TimeStep[[3]]@SplitCany*
*[[1]]*
*[1]    3    5*

Let $C_{im}\epsilon\xi_i$(*first clustering*) be the cluster that experience some external transition and evolve as $C_{jn}\epsilon\xi_j$(*second clustering*). Where the first subscript (i and j) represent time point and second subscript (m and n) represent the cluster number. The *Time Step [3]]* in the output represents the time point $t_j$ at second clustering, and hence the time point $t_i$ ($i = j - 1$) at first clustering $\xi_i$ is one less. So in this particular example $i = 2$ and $j = 3$, then the above transition can be summarized as:

The algorithm detect that three clusters survive ($C_{21}{\rightarrow}C_{31}$, $C_{23}{\rightarrow}C_{34}$, *and* $C_{24}{\rightarrow}C_{32}$) and one cluster split ($C_{22}{\rightarrow}\{C_{33}, C_{35}\}$).

**7.1.2 Example (listdata argument with sliding window model).** In case one is interested in sliding window model, where older records are discarded with the progression of time. This can be achieved by utilizing `swSize` argument. Here in this synthetic example `swSize = 3` will generate window panes that contain datasets arrives in the interval $[t_i - 3 + 1, t_i]$. Analysis of Table 3 demonstrates that the number of clusters in window panes $D_1$, $D_2$, and $D_3$ are 4, 5, and 6 respectively. Hence the *Transition()* function with arguments `listdata = listdata, swSize = 3, typeind = 1, Survival_thrHold = 0.8, Split_thrHold = 0.3,` and `k = c(4,5,6)` can be implemented as:

>*clusterTrace <- Transition(listdata = listdata, swSize = 3, typeind = 1, + Survival_thrHold = 0.8, Split_thrHold = 0.3, k = c(4,5,6))*

## 7.2 Looking at listclus argument

The `listdata` argument permit the users to implement un-clustered datasets $d_1, d_2, \ldots, d_n$ arrives at time-points $t_1, t_2, \ldots, t_n$. However, this restricts the package to only one type of clustering algorithm i.e. k-means algorithm. In order to make the package more flexible for other types of hard clustering, an alternate argument `listclus` is provided in the function. The `listclus` argument imports clustering solutions of each window pane as a list i.e. *listclus =* $\{\xi_1, \xi_2, \ldots, \xi_n\}$ and compute the similarity indices between them. The argument `listclus` is a list, where every individual element is a nested list of matrices or data-frames. The $i^{th}$ element corresponds to the set of clusters $\xi_i = \{X_1, X_2, \ldots, X_{k_i}\}$ extracted at time-point $t_i$, by implementation of an appropriate clustering algorithm to window pane $D_i$. This is explained in the example given below.

**7.2.1 Example: Listclus argument.** Prior to applying *Transition()* function, the user need to extract clusters from each window pane $D_i$. For this purpose, first of all, accumulate the initially collected datasets $d_1, d_2, \ldots, d_n$, according to a suitable window model like landmark in this example. This can be done by explicitly calling *merge()* function from **base** package. By running the R codes given below will generate 4 panes.

>*D1 <- d1*
>*D2 <- merge(d1, d2, all.x = TRUE, all.y = TRUE)*
>*D3 <- merge(D2, d3, all.x = TRUE, all.y = TRUE)*
>*D4 <- merge(D3, d4, all.x = TRUE, all.y = TRUE)*

**Fitting of clustering algorithm**

Afterward, choose the relevant number of clusters from each window pane Di, and extract clusters by implementing an appropriate clustering algorithm. Save this clustering solution as a list of matrices or data frames. For illustration purposes, we obtain 4, 4, 5, and 4 clusters from datasets $D_1$, $D_2$, $D_3$, and $D_4$ respectively.

>*set.seed(100)*
>*fit1 <- kmeans(D1, 4)*
>*C1 <- list()*
>*for(i in 1:4)C1[[i]] <- D1[fit1$cluster == i,]*

where $C1 = \{C_{11}, C_{12}, C_{13}, C_{14}\}$ is a list of clusters extracted from $D_1$ at time point $t_1$. Similarly, extract clusters from all window panes at corresponding time point as:

> *>fit2 <- kmeans(D2, 4)*
> *>C2 <- list()*
> *>for(i in 1:4)C2[[i]] <- D2[fit2$cluster == i,]*
> *>fit3 <- kmeans(D3, 5)*
> *>C3 <- list()*
> *>for(i in 1:5)C3[[i]] <- D3[fit3$cluster == i,]*
> *>fit4 <- kmeans(D4, 4)*
> *>C4 <- list()*
> *>for(i in 1:4)C4[[i]] <- D4[fit4$cluster == i,]*

Combine all these lists of clustering solutions in a single list and apply *Transition()* function with arguments `listclus = listclus, typeind = 3, Survival_thr-Hold = 0.8, Split_thrHold = 0.3` as:

> *>listclus <- list(C1, C2, C3, C4)*
> *>clusterTrace <- Transition(listclus = listclus, typeind = 3,*
> *+ Survival_thrHold = 0.8, Split_thrHold = 0.3)*

## 7.3 Looking at Overlap argument

The `Overlap` argument also permits the user to implement other types of clustering algorithms and trace the evolution of clusters over time. `Overlap` argument imports a list of objects as produced by the *Overlap()* method that contain similarity between clustering obtained at successive time points $t_i$ and $t_j$ ($i < j$) and the summaries of these clusters. This can be implemented by setting `typeind = 2`. The overlap matrices can be computed by utilizing the S4 method *overlap()* exported by the **clusTransition** package. In the same way as `listclus`, some clustering algorithm can be applied to landmark or sliding window modeled dataset to extract the cluster memberships at corresponding time-points. List of clusters extracted from $D_i$ and $D_{i-1}$ can be used to compute the overlap matrix between clustering. This is elaborated in the working example given below.

**7.3.1 Example: Overlap argument.** Let $C1 = \{C_{11}, C_{12}, C_{13}, C_{14}\}$, $C2 = \{C_{21}, C_{22}, C_{23}, C_{24}\}$, $C3 = \{C_{31}, C_{32}, C_{33}, C_{34}, C_{35}\}$, and $C4 = \{C_{41}, C_{42}, C_{43}, C_{44}\}$ be the set of clustering solutions obtained from corresponding datasets $D_1$, $D_2$, $D_3$, and $D_4$. These sets of clustering solutions are already obtained in the previous example. Then the objects of class *OverLap* can be created and initialized as:

> *>obj <- new("OverLap")*
> *>Overlap1 <- Overlap(obj, e1 = C1, e2 = C2)*
> *>Overlap2 <- Overlap(obj, e1 = C2, e2 = C3)*
> *>Overlap3 <- Overlap(obj, e1 = C3, e2 = C4)*

Combine all these objects in a list and apply *Transition()* function with arguments `Overlap = Overlap, typeind = 2, Survival_thrHold = 0.8, Split_thrHold = 0.3` as:

> *>Overlap <- list(Overlap1, Overlap2, Overlap3)*
> *>clusterTrace <- Transition(Overlap = Overlap, typeind = 2,*
> *+ Survival_thrHold = 0.8, Split_thrHold = 0.3)*

## 7.4 *moplot()* function

Fig 3 displays the graphical summary of an object of class **Monic** generated by *Transition()* function as output. The stack bar-plot in the top left corner displays the survival and

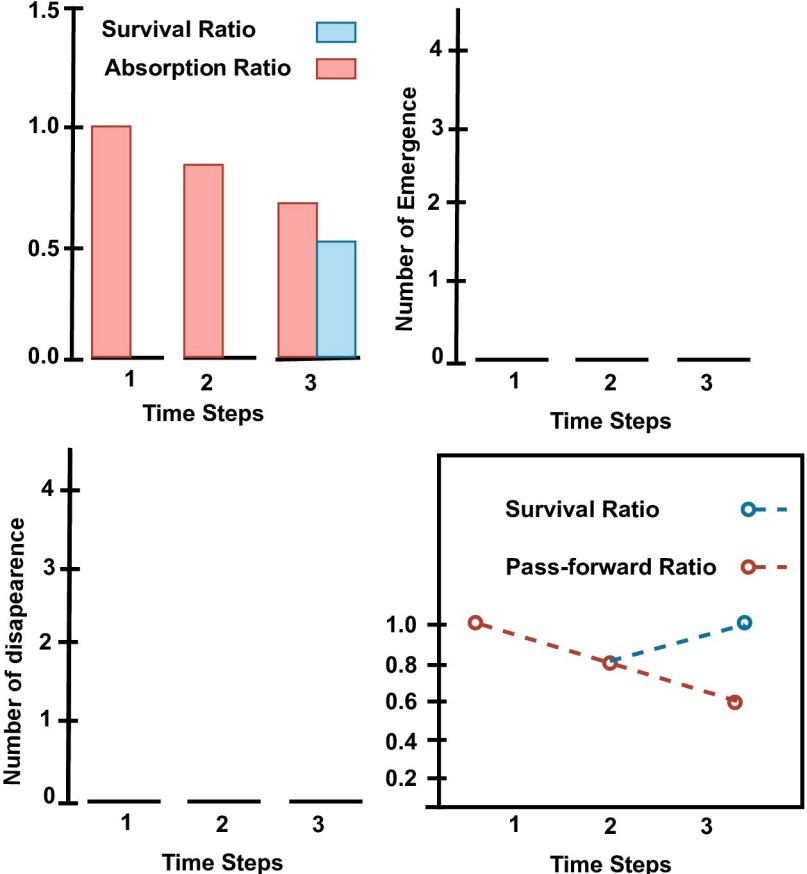

**Fig 3. Data stream generated at four discrete time stamps.** The new data items at each time stamp is shown by the red color, whereas the older data items are shown by black color.

absorption ratio at successive time points. The Figure illustrates that all clusters survived at time point $t_1$, and hence the survival ratio is 1. However, at time point $t_2$ 3 out of 4 clusters survived resulting in a 0.75 survival ratio. Similarly at time point $t_3$ 3 out of 5 clusters survive, while 2 merged. This resulted in 0.60 survival and 0.40 absorption ratios respectively. Consequently, no cluster disappears and no newly emerged candidate were detected at any of the time points. This can be seen from pass-forward ratio, which is unity at all time points except $t_2$ where one cluster splits into daughter candidates.

## 8 Real data example

To demonstrate the practicality of the package and deeply understand applications of cluster evolution, we investigate three real-life datasets. To comprehend the notion of transformation in social, political, and moral attitudes of European nations; the Human Values datasets were extracted from European Social Surveys [35]. The changes in electricity consumption of inhabitants were traced using Individual Household Electricity Consumption dataset. Similarly, the Intel Lab sensors streaming dataset was used to show the applications of the framework. Both these data streams were extracted from the home page of "UCL Machine Learning Repository".

## 8.1 Application to human values scale

As a case study, we extract eight datasets each corresponds to a single round of European Social Surveys (ESS) conducted in years 2002, 2004, 2006, 2008, 2010, 2012, 2014, and 2016 respectively. The dataset consist of 25024 individuals who respond to the Schwartz Value Survey (SVS) for computing basic human values and can be downloaded from the URL https://ess-search.nsd.no/CDW/ConceptVariables. The ten basic values are Benevolence, Universalism, Self-direction, security, Confirmatory, Hedonism, Achievements, Traditions, Stimulation, and Power [35]. The k-means clustering algorithm was implemented to sliding window-modeled datasets at each time point. Whereas, the number of clusters in the respective datasets was estimated from the well-known GAP statistic. Fig 4 below describe the evolution of clusters at time point $t_i$, $i = 1, 2, 3, 4, 5, 6, 7$ in Human Value scale datasets. which demonstrates that two clusters $C_{11}$ and $C_{12}$ survived over time. The first imperative cluster was $C_{11}(C_{11} \rightarrow C_{22} \rightarrow C_{32} \rightarrow C_{42})$ that emerged at $t_1(2002)$ and survived until $t_4(2006, 2010)$. However, the cluster survived till 2010, but experienced internal transition and became more diffused eventually disappeared at time-point $t_5$. The second vibrant cluster was $C_{12}(C_{12} \rightarrow C_{24} \rightarrow C_{33} \rightarrow C_{41} \rightarrow C_{52} \rightarrow C_{63} \rightarrow C_{71})$ which survive through the entire time span. This was the most important cluster because not only it survives over time but also turns out to be denser. Mostly the new respondents of SVS surveys over the years joins this cluster. The shift in location was observed for this cluster at time-point $t_2$ and $t_3$, and afterward, remain stable. The first external transition was experienced in the cluster $C_{14}$ which split into two clusters and ultimately disappeared. The algorithm also detects a cluster $C_{61}$ that emerged at $t_6(2010, 2014)$ and pass-forward while absorbing elements of the cluster $C_{62}$.

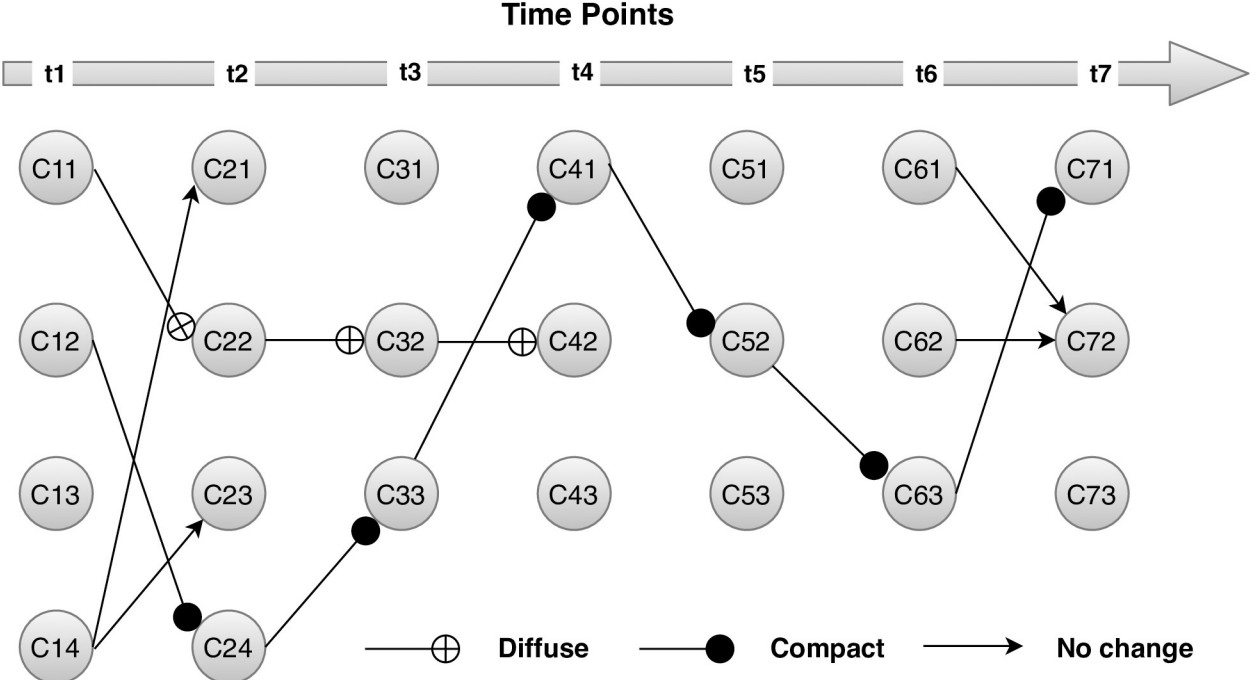

**Fig 4. Transition of clusters in basic human values datasets.**

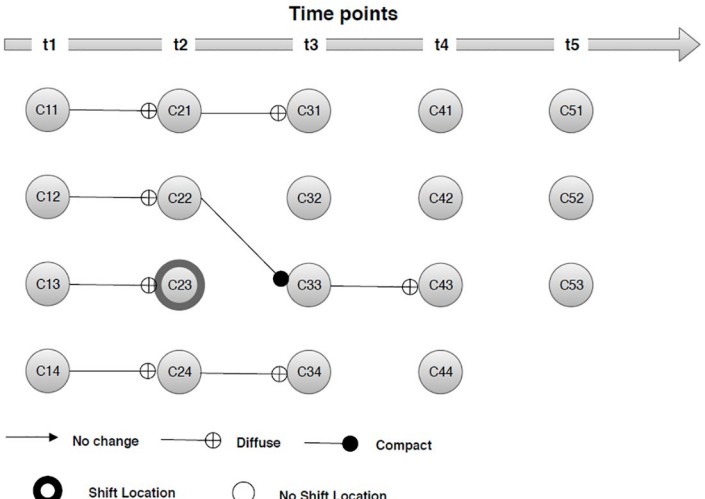

**Fig 5. Transition of clusters in Individual Household Electric Power Consumption datasets.**

## 8.2 Application to Individual Household Electric Power Consumption

As a second example, the Individual Household Electric Power Consumption dataset for the years [2006, 2010] was used. This dataset comprises of 2075259 households characterized by seven numerical attributes. The dataset is available at machine learning repository [36] and can be downloaded from https://archive.ics.uci.edu/ml/datasets/individual+household +electric+power+consumption. A sliding window model of size 2 was used for accumulation of the stream at successive time points. In this section, we use the CLARA algorithm to extract clusters from the datasets at successive time points. Whereas the average silhouette method was used to estimate the optimal value of $k$ in each window pane. Fig 5 below demonstrates the evolution of clusters at time point $t_i$, $i$ = 1, 2, 3, 4, 5 in individual household electric power consumption datasets. The algorithm detect that all of the four clusters survive ($C_{11} \rightarrow C_{21}$, $C_{12} \rightarrow C_{21}$, $C_{13} \rightarrow C_{23}$, and $C_{14} \rightarrow C_{24}$) experiencing internal transition and became diffuse during [2006, 2007]. A shift in location for only one cluster $C_{13}$ was detected, whereas other clusters were stable to change in location. Similarly, three clusters survive ($C_{21} \rightarrow C_{31}$, $C_{22} \rightarrow C_{33}$, and $C_{24} \rightarrow C_{34}$), one cluster disappear ($C_{23} \rightarrow \odot$), and one cluster emerged ($\odot \rightarrow C_{32}$) during [2007, 2008]. Two of the survive clusters became diffuse, while one cluster became compact than its predecessors. Likewise, one cluster survive ($C_{33} \rightarrow C_{43}$), three disappears ($C_{31} \rightarrow \odot$, $C_{32} \rightarrow \odot$, and $C_{34} \rightarrow \odot$), and three newly emerged clusters ($\odot \rightarrow C_{41}$, $\odot \rightarrow C_{42}$, and $\odot \rightarrow C_{44}$) were detected during [2008, 2009]. Afterwards all four clusters disappears ($C_{41} \rightarrow \odot$, $C_{42} \rightarrow \odot$, $C_{43} \rightarrow \odot$, and $C_{44} \rightarrow \odot$), and three new clusters emerged ($\odot \rightarrow C_{51}$, $\odot \rightarrow C_{52}$, and $\odot \rightarrow C_{53}$) during [2009, 2010].

## 8.3 Intel Lab dataset

In this section, we used the publically accessible dataset recorded from 54 sensors deployed at Intel research laboratory during February 28[th] and April 5[th], 2004. Each sensor record information on temperature, humidity, voltage, and light every thirty-one seconds. The dataset comprises of 2.3 million readings collected from 54 sensors. The sensors were designed to make it energy-efficient and consume power only in sensing environment and transmitting data. We select only a subset of measurements from this dataset and include readings from sensor-1 only. This subset of the data consists of 43,047 readings from sensor-1 and can be

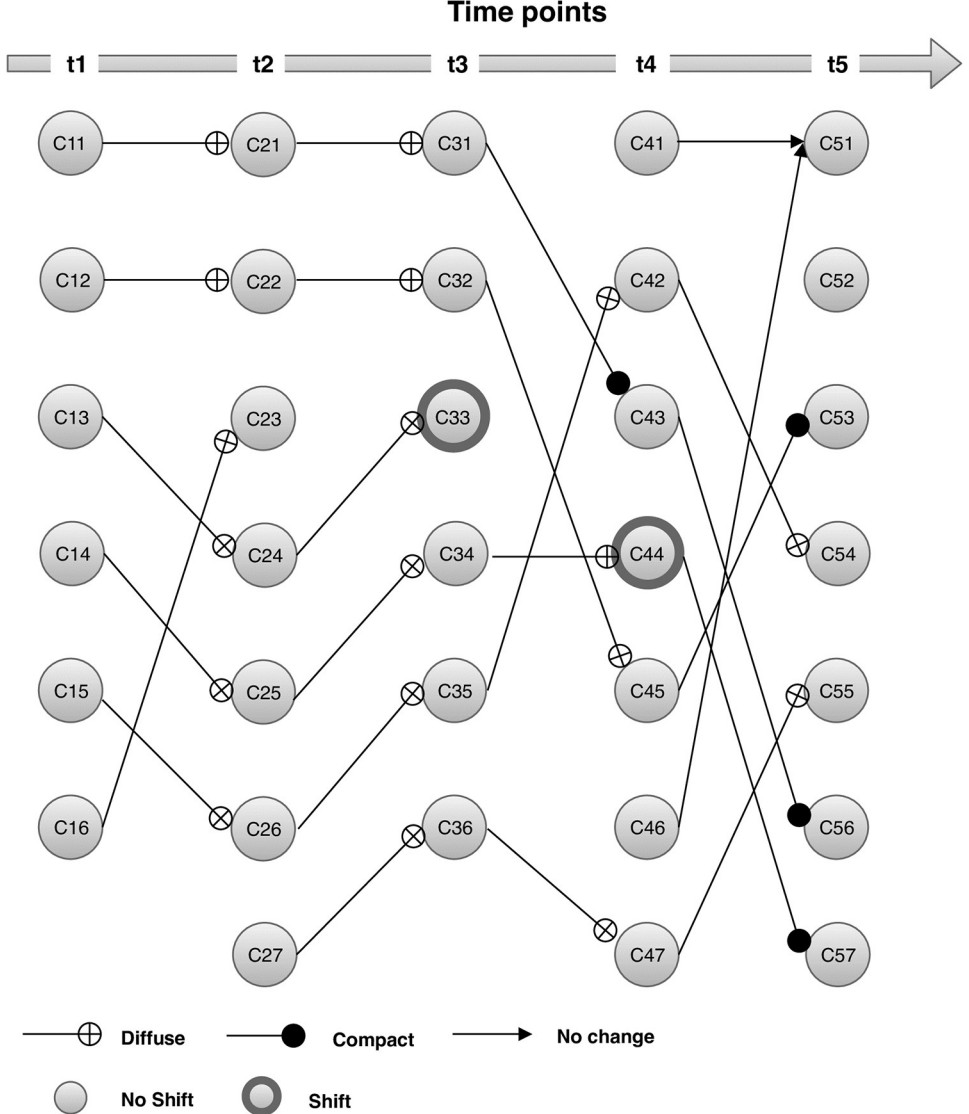

**Fig 6. Evolution of clusters in Intel Lab dataset.**

downloaded from the URL https://www.kaggle.com/datasets/divyansh22/intel-berkeley-research-lab-sensor-data.

We accumulate the dataset according to the landmark window model, and as the flow is uniform, so we consider 9000 records per time period. This implementation generates 5 window panes of cumulative datasets. The shadow statistic decided the optimal number of clusters in cumulative datasets at the corresponding time point. The Partitioned Around Medoids (PAM) algorithm was used for extracting clusters from datasets.

Fig 6 below demonstrates the transitions of clusters at time points $t_i$, $i$ = 1, 2, 3, 4, 5 in Intel Lab dataset. The algorithm detect that all six clusters survive ($C_{11} \rightarrow C_{21}$, $C_{12} \rightarrow C_{22}$, $C_{13} \rightarrow C_{24}$, $C_{14} \rightarrow C_{25}$, $C_{15} \rightarrow C_{26}$, and $C_{16} \rightarrow C_{23}$) while one new cluster emerge ($\odot \rightarrow C_{27}$) at time point $t_2$. All survived clusters experience internal transition and became more diffuse. Also six clusters survive ($C_{21} \rightarrow C_{31}$, $C_{22} \rightarrow C_{32}$, $C_{24} \rightarrow C_{33}$, $C_{25} \rightarrow C_{34}$, $C_{26} \rightarrow C_{35}$, and $C_{27} \rightarrow C_{36}$) and one cluster disappears ($C_{23} \rightarrow \odot$) at time point $t_3$. Cluster $C_{24}$ experience double internal transition i.e.

shift in location and change in density, while other clusters only became diffuse. Likewise, five clusters survive ($C_{31} \rightarrow C_{43}$, $C_{32} \rightarrow C_{45}$, $C_{34} \rightarrow C_{44}$, $C_{35} \rightarrow C_{42}$, and $C_{36} \rightarrow C_{47}$), one cluster disappears ($C_{33} \rightarrow \odot$), and two clusters emerged ($\odot \rightarrow C_{41}$ and $\odot \rightarrow C_{46}$) at time point $t_4$. Similarly, five clusters survive ($C_{42} \rightarrow C_{54}$, $C_{43} \rightarrow C_{56}$, $C_{44} \rightarrow C_{57}$, $C_{45} \rightarrow C_{53}$, and $C_{47} \rightarrow C_{55}$), two clusters merge ($\{C_{41}, C_{46}\} \rightarrow C_{51}$), whereas one cluster emerge ($\odot \rightarrow C_{52}$) at time point $t_5$.

For further details and understanding the significance and practical applications of monitoring changes in clustering solutions of streaming datasets see Atif et al [37].

## 9 Concluding remarks

In this paper, we introduce an R package clusTransition dedicated to trace the evolution of cluster solutions in cumulative datasets. The package implements state-of-the-art algorithm **MONIC** for modeling and tracing the transition of cluster solutions in dynamic datasets. This algorithm is based on re-clustering of cumulative datasets $D_1, D_2, \ldots, D_n$ arriving at corresponding time-points $t_1, t_2, \ldots, t_n$ and monitor the changes occurring in these cluster solutions. The changes comprise of clusters that still exist, split into various, absorbed by others, disappeared and newly emerged. The clusters that survived in external transition may experience a change in location and density called internal transition. We have applied **clusTransition** package on synthetic as well as on real-life datasets to look insight into change detection framework.

## 10 Limitations of the package

The clusTransition package takes into account batch processing, where the stream is discretized and the gathered data is put into the windowing model. The datasets are not clustered upon arrival immediately in real time. Similarly, the use of sliding and landmark models either contain the data items or entirely ignore them at subsequent time-points. A damped window model, on the other hand, assigns each object, depending on its arrival time, exponentially decreasing weights. Future plans call for adding support for the damped window model to the R package for change detection.

The paradigm for cluster transition monitoring presupposes hard clustering, which requires that each item be assigned to one and only one cluster. This assumption implies that the strategy cannot be used to density-based or model-based clustering approaches, leaving the problem open for further investigation.

## Author Contributions

**Conceptualization:** Muhammad Atif, Friedrich Leisch.

**Data curation:** Muhammad Atif.

**Formal analysis:** Muhammad Atif.

**Methodology:** Muhammad Atif.

**Project administration:** Friedrich Leisch.

**Software:** Muhammad Atif, Friedrich Leisch.

**Supervision:** Friedrich Leisch.

**Validation:** Friedrich Leisch.

**Writing – original draft:** Muhammad Atif.

**Writing – review & editing:** Friedrich Leisch.

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
