## [Decision Letter · Decision Letter 0]

12 Aug 2022

PONE-D-22-20071clusTransition: An R Package for Monitoring Transition in Cluster Solutions of Temporal DatasetsPLOS ONE

Dear Dr. Atif,

Thank you for submitting your manuscript to PLOS ONE. After careful consideration, we feel that it has merit but does not fully meet PLOS ONE’s publication criteria as it currently stands. Therefore, we invite you to submit a revised version of the manuscript that addresses the points raised during the review process.

We look forward to receiving your revised manuscript.

Kind regards,

Mohammad Mehdi Rashidi

Academic Editor

PLOS ONE

Journal Requirements:

Reviewers' comments:

Reviewer's Responses to Questions

**Comments to the Author**

1. Is the manuscript technically sound, and do the data support the conclusions?

Reviewer #1: Yes

Reviewer #2: Yes

2. Has the statistical analysis been performed appropriately and rigorously? 

Reviewer #1: Yes

Reviewer #2: N/A

3. Have the authors made all data underlying the findings in their manuscript fully available?

Reviewer #1: No

Reviewer #2: Yes

4. Is the manuscript presented in an intelligible fashion and written in standard English?

Reviewer #1: Yes

Reviewer #2: Yes

5. Review Comments to the Author

Reviewer #1: This paper proposes a R package that implementing MONIC framework for clustering temporal datasets. It has some merits to be published. However, it still has some problems.

1. The workflow of the proposed clusTransition should be given;

2. What are the major limitations of this R package?

3. More comparative experiments are required.

4. More important articles should be cited.

5. More standard data sets from UCI should be checked.

Reviewer #2: In this paper, an R package called clusTransition is discussed, and the use of the package is demonstrated. I found the package interesting. However, the presentation needs improvements. I listed my comments below:

1. There is no need to menaiton t1, t2, ..., tn in the abstract. Please consider removing it.

2. In the Abstract, it is mentioned that "The contribution of this paper is to demonstrate the implementation of the package using synthetic and real-life datasets in R software." This sentence gives the impression that this paper is demonstrating someone else's package. However, the package's author is the first author of the manuscript. Please revise this sentence.

3. Please do not give mathematical definitions in the Introduction. Instead, you should discuss the necessity of the package and such a paper to describe it, mention the contributions and finish with an outline paragraph. You can move the details of the method that the package implements to the next section.

4. Please link the function descriptions of the package to the mathematical definition of the methods implemented by the package.

5. It looks like the help documentation of the package is repeated in the manuscript. Since all those points are already given in the documentation, repeating them is unnecessary. Instead, please discuss how to specify the inputs and how to use the outputs in relation to the methods.

6. Manuscript needs to be checked against English language issues.

7. It is mentioned that "we generate a data stream sprouting at four consecutive time points". Please elaborate on the generation of data.

8, It is mentioned that "... we generate a data stream sprouting at four consecutive time points." But in the next sections, there are only two applications: "Application to Human values scale" and "Application to Individual Household Electric Power Consumption."

6. PLOS authors have the option to publish the peer review history of their article (what does this mean?). If published, this will include your full peer review and any attached files.

Reviewer #1: No

Reviewer #2: No

---

## [Author Response · Author response to Decision Letter 0]

10 Sep 2022

Thank you for giving me the opportunity to submit a revised draft of my manuscript titled clusTransition: An R Package for Monitoring Transition in Cluster Solutions of Temporal Datasets to PLOS ONE. We appreciate the time and effort that you and the reviewers have dedicated to providing your valuable feedback on my manuscript. We are grateful to the reviewers for their insightful comments on my paper. We have highlighted the changes within the manuscript.

Here is a point-by-point response to the reviewers’ comments and concerns.

Comments from Reviewer # 1: 

Comment # 1: The workflow of the proposed clusTransition should be given;

Response: Added to the manuscript with track changes.

Comment # 2: What are the major limitations of this R package?

Response: Added to the manuscript with track changes. 

Comment # 3: More comparative experiments are required.

Response: Most of the articles that discuss analysis of streaming data use real-life datasets. Unfortunately, there is no guidance available for conducting comparative experiments. However, Atif. (2021) perform a simulated study to analyze the performance of framework for various clustering parameters.

Comment # 4: More important articles should be cited.

Response: Added to the manuscript with track changes.

Comments # 5: More standard data sets from UCI should be checked.

Response: Thank you for pointing out this, I agree with this comment. Therefore I have added some standard datasets from UCL machine learning repository. So we added the Intel Berkeley Research Lab Sensor Data to the manuscript used by many research articles such as Doreswamy, Narasegouda, S. (2014), Baralis, Cerquitelli and D'Elia (2007), Baralis, Elena & Cerquitelli, Tania & D’Elia, Vincenzo. (2022) etc.

Furthermore I added a citation of the paper that discusses the applications of monitoring changes in clustering solutions using some standard datasets.

 Comments from Reviewer # 2: 

Comment # 1: There is no need to mention t1, t2, ..., tn in the abstract. Please consider removing it.

Response: Corrected accordingly. 

Comment # 2: In the Abstract, it is mentioned that "The contribution of this paper is to demonstrate the implementation of the package using synthetic and real-life datasets in R software." This sentence gives the impression that this paper is demonstrating someone else's package. However, the package's author is the first author of the manuscript. Please revise this sentence.

Response: The statement is rephrased in the manuscript with track changes. 

Comment # 3: Please do not give mathematical definitions in the Introduction. Instead, you should discuss the necessity of the package and such a paper to describe it, mention the contributions and finish with an outline paragraph. You can move the details of the method that the package implements to the next section.

Response: All the mathematical definitions and methods used in the package are discussed in the section “Change detection algorithms”. These definitions and mathematical descriptions are removed from the “Introduction” section in the updated manuscript. 

Comment # 4: Please link the function descriptions of the package to the mathematical definition of the methods implemented by the package.

Response: Corrected accordingly.

Comments # 5: It looks like the help documentation of the package is repeated in the manuscript. Since all those points are already given in the documentation, repeating them is unnecessary. Instead, please discuss how to specify the inputs and how to use the outputs in relation to the methods.

Response: Corrected in the manuscript with track changes. 

Comment # 6: Manuscript needs to be checked against English language issues.

Response: Corrected in the manuscript with track changes. 

Comment # 7: It is mentioned that "we generate a data stream sprouting at four consecutive time points". Please elaborate on the generation of data.

Response: The package takes temporal datasets as input and re-clusters them at successive time points. The temporal datasets is not stationary, rather it evolve over time. The temporal datasets have a dedicated attribute known as time-stamp, which record the arrival time of each data record. The stream of data records is discritize by accumulating it at discrete time points denoted by t1, t2, ..., tn respectively. 

Comments # 8: It is mentioned that "... we generate a data stream sprouting at four consecutive time points." But in the next sections, there are only two applications: "Application to Human values scale" and "Application to Individual Household Electric Power Consumption."

Response: We have generated a synthetic temporal dataset that evolve at four consecutive time points. The four time points refer to the time-stamps at which data records emerged having same attributes. The does not indicate different datasets. The applications i.e. “Application to Human values scale" and "Application to Individual Household Electric Power Consumption" are also temporal datasets which emerged at 5 time-stamps. 

• Doreswamy, Narasegouda, S. (2014). Fault Detection in Sensor Network Using DBSCAN and Statistical Models. In: Satapathy, S., Udgata, S., Biswal, B. (eds) Proceedings of the International Conference on Frontiers of Intelligent Computing: Theory and Applications (FICTA) 2013. Advances in Intelligent Systems and Computing, vol 247. Springer, Cham. https://doi.org/10.1007/978-3-319-02931-3_50

• E. Baralis, T. Cerquitelli and V. D'Elia, "Modeling a Sensor Network by means of Clustering," 18th International Workshop on Database and Expert Systems Applications (DEXA 2007), 2007, pp. 177-181, doi: 10.1109/DEXA.2007.23.

• Baralis, Elena & Cerquitelli, Tania & D’Elia, Vincenzo. (2022). Technical Report Modeling a Sensor Network by means of Clustering.

• M. Atif. (2021). Monitoring changes in cluster solutions. (Doctoral dissertation). Available from FIS of University of Natural Resources and Life Sciences, Vienna.

---

## [Decision Letter · Decision Letter 1]

12 Oct 2022

PONE-D-22-20071R1clusTransition: An R Package for Monitoring Transition in Cluster Solutions of Temporal DatasetsPLOS ONE

Dear Dr. Atif,

Thank you for submitting your manuscript to PLOS ONE. After careful consideration, we feel that it has merit but does not fully meet PLOS ONE’s publication criteria as it currently stands. Therefore, we invite you to submit a revised version of the manuscript that addresses the points raised during the review process.

We look forward to receiving your revised manuscript.

Kind regards,

Mohammad Mehdi Rashidi

Academic Editor

PLOS ONE

Journal Requirements:

Reviewers' comments:

Reviewer's Responses to Questions

**Comments to the Author**

1. If the authors have adequately addressed your comments raised in a previous round of review and you feel that this manuscript is now acceptable for publication, you may indicate that here to bypass the “Comments to the Author” section, enter your conflict of interest statement in the “Confidential to Editor” section, and submit your "Accept" recommendation.

Reviewer #2: All comments have been addressed

2. Is the manuscript technically sound, and do the data support the conclusions?

Reviewer #2: Yes

3. Has the statistical analysis been performed appropriately and rigorously? 

Reviewer #2: Yes

4. Have the authors made all data underlying the findings in their manuscript fully available?

Reviewer #2: Yes

5. Is the manuscript presented in an intelligible fashion and written in standard English?

Reviewer #2: Yes

6. Review Comments to the Author

Reviewer #2: The author has responded to all my comments in the previous review round sufficiently. I have one more minor comment. Please consider merging the "Related works" section into the introduction since this section is too small to be a stand-alone section in the manuscript.

7. PLOS authors have the option to publish the peer review history of their article (what does this mean?). If published, this will include your full peer review and any attached files.

Reviewer #2: No

---

## [Author Response · Author response to Decision Letter 1]

15 Oct 2022

Reviewer #2: The author has responded to all my comments in the previous review round sufficiently. I have one more minor comment. Please consider merging the "Related works" section into the introduction since this section is too small to be a stand-alone section in the manuscript.

Response: The Related work section is merge with the Introduction section. The changes are highlighted in revised manuscript with track changes.

---

## [Decision Letter · Decision Letter 2]

11 Nov 2022

clusTransition: An R Package for Monitoring Transition in Cluster Solutions of Temporal Datasets

PONE-D-22-20071R2

Dear Dr. Atif,

We’re pleased to inform you that your manuscript has been judged scientifically suitable for publication and will be formally accepted for publication once it meets all outstanding technical requirements.

Kind regards,

Mohammad Mehdi Rashidi

Academic Editor

PLOS ONE

Additional Editor Comments (optional):

Reviewers' comments:

Reviewer's Responses to Questions

**Comments to the Author**

1. If the authors have adequately addressed your comments raised in a previous round of review and you feel that this manuscript is now acceptable for publication, you may indicate that here to bypass the “Comments to the Author” section, enter your conflict of interest statement in the “Confidential to Editor” section, and submit your "Accept" recommendation.

Reviewer #2: All comments have been addressed

2. Is the manuscript technically sound, and do the data support the conclusions?

Reviewer #2: Yes

3. Has the statistical analysis been performed appropriately and rigorously? 

Reviewer #2: Yes

4. Have the authors made all data underlying the findings in their manuscript fully available?

Reviewer #2: Yes

5. Is the manuscript presented in an intelligible fashion and written in standard English?

Reviewer #2: Yes

6. Review Comments to the Author

Reviewer #2: In this revision round, the author has responded to all my comments in the previous review round sufficiently.

7. PLOS authors have the option to publish the peer review history of their article (what does this mean?). If published, this will include your full peer review and any attached files.

Reviewer #2: No

---

## [Editor Report · Acceptance letter]

17 Nov 2022

PONE-D-22-20071R2 

clusTransition: An R Package for Monitoring Transition in Cluster Solutions of Temporal Datasets 

Dear Dr. Atif:

I'm pleased to inform you that your manuscript has been deemed suitable for publication in PLOS ONE. Congratulations! Your manuscript is now with our production department. 

Kind regards, 

on behalf of

Professor Mohammad Mehdi Rashidi 

Academic Editor

PLOS ONE